# Improving case-detection of wasting among under-five-year-old children in Ethiopia: A secondary analysis of community-based surveys in humanitarian settings

Alinoor Mohamed Farah[1,2], Hamid Yimam Hassen[3,4], Sibhatu Biadgilign[1], Aweke Kebede[5], Yakob Desalegn[1], Beza Yilma[1], Tesfamichael Awoke[1], Samson Gebremedhin[1], Kemeria Barsenga[5], Tafara Ndumiyana[5], Robert Ackatia-Armah[5], Helina Tufa[6], Firaol Bekele[6], Hailu Wondim[7], Eskeziaw Abebe[1], Seifu Hagos Gebreyesus[1]*

**1** School of Public Health, College of Health Sciences, Addis Ababa University, Addis Ababa, Ethiopia, **2** Department of Public Health Nutrition, School of Public Health, College of Medicine and Health Sciences, Jigjiga University, Jigjiga, Ethiopia, **3** Department of Primary & Integrated Care, Department of Family Medicine and Population Health, Faculty of Medicine and Health Sciences, University of Antwerp, Antwerp, Belgium, **4** Environmental Intelligence Unit, Flemish Institute for Technological Research (VITO), Mol, Belgium, **5** World Food Programme (WFP), Addis Ababa, Ethiopia, **6** Federal Disaster Risk Management Commission, Ethiopia and Emergency Nutrition Coordination Unit, Addis Ababa, **7** Action Against Hunger Canada, Global SMART Initiative, Toronto

* seif_h23@yahoo.com

## Abstract

WHO recommends weight-for-height Z-score (WHZ) <-3 or Mid-Upper Arm Circumference (MUAC) < 125 mm as criteria for diagnosing wasting in children aged 6–59 months. In humanitarian settings, MUAC provides a simpler alternative than WHZ measurements, requiring only a tape measure. However, using MUAC alone may miss many at-risk children, causing discrepancies in wasting estimates. We analyzed 31 Standardized Monitoring and Assessment of Relief and Transitions (SMART) surveys from 2022-2025 across eight Ethiopian regions. The sample included 23,419 children aged 6–59 months with complete data. MUAC's diagnostic performance for GAM was evaluated using WHZ as reference, applying standard parameters including sensitivity, specificity, PPV, NPV, Youden index, and ROC curve analyses. We assessed MUAC cut-offs' accuracy nationally by age group (6–23 and 24–59 months) and administrative region. The optimal diagnostic MUAC cutoff for WHZ<−2 was 139 mm, with a Youden Index of 43%. This cutoff varied by age: 128 mm for children under 24 months (Youden Index: 54.5%) and 140 mm for children 24 months and older (Youden Index: 42.5%). MUAC diagnostic performance varied across regions, with optimal thresholds from 126.5 mm in Tigray to 143.5 mm in Gambella. Global acute malnutrition (GAM) prevalence differs based on the indicator used (MUAC versus WHZ-based), with MUAC sometimes overestimating or underestimating GAM compared to WHZ. Increasing the MUAC cutoff from 125 mm to 139 mm improves

**Data availability statement:** The data set for this study can be found in figshare 10.6084/m9.figshare.31375354.

**Funding:** The authors received no specific funding for this work.

**Competing interests:** The authors have declared that no competing interests exist.

case detection but substantially increases false positives, showing a sensitivity-specificity trade-off. A single universal MUAC cutoff may under- and overestimate global acute malnutrition. Increasing the cutoff to 139 mm improves detection but raises false positives, highlighting the trade-off between sensitivity and specificity and emphasizing the need for context-specific thresholds to optimize screening. However, implementing context-specific thresholds remains uncertain.

## Introduction

Childhood malnutrition remains a major global health challenge, affecting millions of children under five years of age in low- and middle-income countries [1–3]. Wasting, defined as low weight-for-height, indicates acute malnutrition among children under five [3], resulting from poor dietary intake, infections, and socioeconomic conditions, leading to developmental delays and increased mortality risk when severe [2]. According to the World Health Organization (WHO) and United Nations Children's Fund (UNICEF) (2023), wasting affected 6.8% or 45 million children under 5 years of age globally in 2022, with 13.6 million (2.1%) suffering severe wasting. Over 75% of children with severe wasting live in Asia and 22% in Africa [2]. Children with wasting face a higher mortality risk, especially when they are stunted or underweight [4]. The global community's Sustainable Development Goals (SDGs) aim to reduce wasting to <5% by 2025 and <3% by 2030 [5]. According to the Ethiopia Mini-Demographic and Health Survey (EMDHS) 2019, 7% of Ethiopian children are wasted. The highest prevalence of wasted children is in Somali (21%), Afar (14%), and Gambella (13%), while the lowest is in Addis Ababa (2%) and Harari (4%) [6].

Early identification and treatment of wasting in children are critical for improving health outcomes and reducing mortality among vulnerable populations [7–9]. Wasting can be categorized as severe, with a weight-for-height Z-score (WHZ) <-3 or mid-upper arm circumference <115 mm, or moderate, with WHZ between -2 and -3 or mid-upper arm circumference (MUAC) between 115 and 125 mm [10,11]. Global Acute Malnutrition (GAM) includes both moderate and severe forms and thresholds are WHZ<−2 or MUAC<125 mm [12]. Although MUAC is a more effective predictor of mortality in both community [13,14] and hospitalized settings [15], and can be effectively implemented by minimally trained personnel [16].. Moreover, the current WHO-recommended MUAC cutoffs (115 mm for severe and 125 mm for moderate wasting) have limited sensitivity, identifying only 6–20% of severe cases and 13–23% of moderate cases across different settings [17].

Compounding these measurement challenges, the concordance between MUAC and WHZ in identifying wasted children varies substantially across studies. While some studies have found moderate agreement between MUAC and WHZ [18,19], others have reported poor concordance [20,21]. MUAC and WHZ often identify different sets of malnourished children. MUAC tends to be more sensitive in identifying younger children and girls. These differences likely stem from the fact that MUAC uses a single cut-off to define wasting and may over diagnose the condition in

some sub-populations while underestimating cases in older children, particularly boys [22]. WHZ-based diagnosis would be less sensitive to this bias as it standardizes for sex and height, thus age [22]. Some studies have explored MUAC-for-age (MUACZ) as an alternative, finding improved agreement with WHZ compared with MUAC alone [23,24]. However, the benefits of MUACZ remain inconclusive. Given the discrepancies between MUAC and WHZ, some researchers have suggested using both criteria for comprehensive malnutrition screening [25], especially in areas with a high prevalence of chronic malnutrition and kwashiorkor [24].

In fragile humanitarian contexts, where precise WHZ measurements require specialized equipment and trained person-nel, MUAC's simplicity (requiring only a tape measure) makes it invaluable for rapid screening [17,26]. However, relying on MUAC alone risks missing many at-risk children and underestimating the prevalence of wasting, thereby increasing mortality from undiagnosed cases [17,26,27]. In Ethiopia, health posts commonly use MUAC as the sole criterion for iden-tifying and managing wasting because of logistical challenges in obtaining accurate length/height data. However, given that MUAC and WHZ classify different groups of children as wasted, with misclassification rates that fluctuate by region and population, the relationship between GAM by MUAC and GAM by WHZ can diverge markedly. Therefore, this study aimed to evaluate the diagnostic performance of MUAC in identifying acute malnutrition among Ethiopian children aged 6–59 months nationally and document variation across age groups and among regions in Ethiopia.

## Methods

### Source of data

The study drew on 31 livelihood zones and district-level surveys conducted between 2022 and 2025 across eight regions: Amhara, Afar, Benishangul-Gumuz, Gambella, Oromia, Somali, Southern and Tigray. All the datasets were obtained on August 01, 2025. All surveys were population-based and designed to be representative of the livelihood zone and district level, following the Standardized Monitoring and Assessment of Relief and Transitions (SMART) methodology. They were implemented to assess the need for emergency nutrition programs, led by the Government through the Emergency Nutri-tion Coordination Unit (ENCU) in collaboration with the implementing partners.

A two-stage cluster sampling approach was applied in all surveys, with probability proportional to population size used in the first stage, consistent with the SMART methodology. Anthropometric measurements were collected using standard-ized instruments. Children's height was measured using UNICEF height/length boards with a precision of 1mm, while weight was measured using Seca scales and recorded to the nearest 0.1 kg.

Enumerators underwent training and a standardization test to ensure measurement quality. The SMART methodology includes standardized enumerator training modules that typically last five days. During this period, enumerators receive both theoretical and practical sessions and complete a standardization test to ensure they can accurately measure chil-dren. If enumerators fail the test, the survey cannot proceed until they retake and pass it. The training also includes a pilot phase, where enumerators practice conducting the survey before the actual data collection begins. For children without official documentation of birth dates, a local events calendar was used to estimate their age in months.

### Data processing

Raw datasets from anthropometric surveys, including information on age, sex, weight, height/length and MUAC, were obtained from the SMART+ aggregator, a repository where non-governmental organizations (NGOs), United Nations (UN) agencies, and governments publish SMART+ survey data with prior approval from the Ethiopian Disaster Risk Manage-ment Commission (EDRMC), and Ethiopia Emergency Nutrition Coordination Unit (ENCU). Data cleaning was conducted using the Emergency Nutrition Assessment (ENA) for the SMART software (version 2020). Z-scores were calculated according to the WHO 2006 growth reference, and SMART flags (±3 z-scores) were applied to exclude implausible WHZ values from the analyses. The overall quality of the datasets was verified using the ENA plausibility check to ensure com-pliance with SMART quality standards. MUAC data were retained without exclusion criteria.

Following data cleaning, the datasets were imported into R software (version 4.5.1) for further analysis. Nutritional status was classified according to the standard definitions of the WHO. Wasting by WHZ was defined as <−2 z-scores and wasting by MUAC as <125 mm. Global Acute Malnutrition (GAM) was defined as WHZ <−2.0 or MUAC <125 mm. Severe Acute Malnutrition (SAM) was classified as WHZ <−3.0 or MUAC <115 mm. Moderate Acute Malnutrition (MAM) was defined as WHZ between −3.0 and −2.0 or MUAC between 115 mm and <125 mm. Cases of bilateral pitting edema were not included in estimated prevalence of wasting by WHZ or MUAC; edema cases were relatively rare, representing approximately 0.05% of the total cases. WHZ was missing for 1,108 children, 1,095 cases with missing height or weight, and 13 cases with missing age. Thus, all WHZ-missing records were excluded because the required inputs for the WHZ calculation were unavailable.

## Statistical analysis

Data were analyzed to assess the comparability and diagnostic performance of MUAC compared with WHZ in identifying acute malnutrition across regions and age groups. Descriptive statistics (means, standard deviations, and proportions) were calculated for anthropometric indicators WHZ, MUAC, Height-for-Age Z-score (HAZ), and Weight-for-Age Z-score (WAZ), stratified by region. Group differences were examined by comparing the mean values and prevalence estimates.

The relationship between MUAC and WHZ was evaluated using linear regression models, with WHZ as the dependent variable and MUAC as the predictor. Scatter plots and region-specific regression coefficients were used to illustrate the strength and direction of the associations. Goodness-of-fit was assessed using the coefficient of determination ($R^2$), and slopes were compared across regions to evaluate variability in predictive strength.

The prevalence of MAM and SAM was estimated using the WHO standard cut-offs for both MUAC and WHZ. Agreement between MUAC- and WHZ-based classifications was assessed using cross-tabulations, inflation factors (ratio of WHZ-based to MUAC-based caseloads), and Cohen's kappa statistics. Kappa statistics were used to assess the level of agreement between MUAC and WHZ measurements in identifying children classified as wasted. Cohen proposed that Kappa values should be understood in the following way: values ≤ 0 reflects no agreement; 0.01–0.20 suggest none to slight, 0.21–0.40 as fair agreement, 0.41– 0.60 represents a moderate level of agreement, 0.61–0.80 as substantial, and 0.81–1.00 signals an almost perfect agreement [28,29]. Pearson's correlation coefficients were calculated to evaluate the strength and direction of the linear relationship between MUAC and WHZ, thereby assessing the correlation between the anthropometric indicators. The strength of Pearson's correlation coefficient is commonly interpreted using specific ranges. Values close to ±1 indicate a perfect correlation, showing that the variables move together in the same or opposite direction. Coefficients between ±0.50 and ±0.99 were considered high, reflecting a strong relationship. Values from ±0.30 to ±0.49 suggest a moderate association, indicating a medium level of connection. Values below ±0.29 imply a weak or low relationship, while a value of zero indicates no correlation, meaning there is no linear relationship between the variables [29,30].

The diagnostic performance of MUAC in identifying WHZ-defined wasting was examined using receiver operating characteristic (ROC) curve analysis, with area under the curve (AUC) values computed for each region, enabling the identification of variations in diagnostic accuracy across specific subpopulations. The area under the curve (AUC) was used as a measure of diagnostic accuracy, ranging from 0.5 (no better than chance) to 1 (perfect accuracy). Optimal cut-offs were determined using Youden's index (Sensitivity+Specificity−1). This index ranges from 0 to 1, with higher values indicating better diagnostic performance. The MUAC cutoff with the highest Youden's Index was selected as the optimal threshold because it maximized the combined sensitivity and specificity. Sensitivity, specificity, positive predictive value (PPV), negative predictive value (NPV), accuracy, and likelihood ratios ($LR^+$, $LR^-$) were calculated for each cutoff value. Sensitivity was defined as the proportion of wasted children (based on the gold standard WHZ) correctly identified by the MUAC cutoff, and specificity was defined as the proportion of non-wasted children correctly classified as not wasted. The positive predictive value (PPV) represents the likelihood that children classified as wasted by MUAC truly have wasting according

to WHZ, whereas the negative predictive value (NPV) represents the likelihood that those classified as not wasted by MUAC truly do not have wasting by WHZ. Accuracy reflects the overall proportion of children correctly classified as wasted or not wasted when MUAC was compared with WHZ. Likelihood ratios were also calculated: the positive likelihood ratio (LR+) indicates how much more likely a positive MUAC result is in children with wasting by WHZ compared to those without, and the negative likelihood ratio (LR-) indicates how much less likely a negative MUAC result is in children with wasting compared to those without. Analyses were stratified by region and age group (<24 and ≥24 months) to capture the heterogeneity in MUAC performance. False-positive rates (FPR) and false-negative rates (FNR) were estimated across a range of MUAC thresholds to quantify misclassification risks when using MUAC compared to WHZ. Regional variation in FPR/FNR profiles was highlighted to identify settings in which MUAC underestimates or overestimates the burden of wasting. All analyses were conducted using R (version 4.4.1).

## Ethics statement

This was a secondary analysis of anonymous data in which no individual cluster or village location could be identified; therefore, formal ethical clearance was not required. Permission to use and analyze the dataset was obtained from the EDRMC and ENCU, which provided raw datasets.

## Result

### Characteristics of surveys and study population

As shown in Table 1, 31 SMART+ surveys were included, covering 23,419 children across eight Ethiopian regions. The highest number of surveys was conducted in the Somali Region (10 surveys, 7,341 children), followed by Oromia (6 surveys, 4,571 children), Amhara (5 surveys, 3,578 children), and Tigray (5 surveys, 3,033 children). Fewer surveys were conducted in Afar (two surveys, 1,483 children), Benishangul-Gumuz (one survey, 537 children), Gambella (one survey, 1,140 children), and Southern Ethiopia (one survey, 1,736 children). The median age of the participants was 32 months (IQR: 20–46). The sample was nearly evenly split by sex, with 11,867 (51%) boys and 11,552(49%) girls.

Table 1. Distribution of surveys and study participants by region.

| Characteristics | N | N |
|---|---|---|
| Regions | Surveys | Children |
| Tigray | 5 | 3033 |
| Afar | 2 | 1483 |
| Amhara | 5 | 3578 |
| Oromia | 6 | 4571 |
| Somali | 10 | 7341 |
| Beshangul-Gumuz | 1 | 537 |
| Gambella | 1 | 1140 |
| Southern Ethiopia | 1 | 1736 |
| Total | 31 | 23419 |
| ¹Age (months) | | 32 (20, 46) |
| Sex | | |
| Boys | 51% | 11867 |
| Girls | 49% | 11552 |

¹n (%); Median (IQR).

## Descriptive summary of anthropometric indicators by region

As presented in Table 2, the overall mean WHZ across all eight regions was –0.81, with regional variation. The lowest WHZ was recorded in Somali (–1.02), followed by Afar (–0.97) and Tigray (–0.87). The highest WHZ was found in Oromia (–0.48).

The overall mean MUAC was 142.09 mm. The lowest regional mean MUAC was reported in Tigray (134.31 mm), whereas the highest was reported in Gambella (146.55 mm). Somali, despite recording the lowest WHZ, had a relatively high mean MUAC (145.83 mm).

## Relationship between MUAC and WHZ by region

The histograms of WHZ by region (Fig 1) show approximately normal distributions, with Somali and Tigray displaying left-skewed patterns that indicate lower WHZ values than the other regions. MUAC distributions (Fig 2) are also roughly normal, with most regions concentrated around 140–150 mm. Somali and Gambella show broader MUAC spreads, whereas Tigray exhibits a left skewed distribution. Both measures revealed regional variations in the acute nutritional status of children across Ethiopia.

The scatter plots of MUAC versus WHZ by region (Fig 3) demonstrate a positive linear relationship across all regions, suggesting that higher MUAC values are generally associated with higher WHZ. The strength of this relationship varies by region. Benishangul-Gumuz exhibits the strongest correlation ($R^2 = 0.43$), followed by Tigray, Afar, Amhara, and South Ethiopia, which display moderate correlations ($R^2 = 0.32$–0.36). Oromia presented a weaker correlation ($R^2 = 0.29$), whereas Somali and Gambella showed the lowest correlations ($R^2 = 0.23$ and 0.21, respectively). The regression slopes differed across regions. Tigray and Benishangul-Gumuz exhibited steeper slopes (0.0566), indicating that small changes in MUAC corresponded to larger changes in WHZ than in other regions.

## Nutritional status of children by WHZ and MUAC indicators

Table 3 presents the regional distribution of wasting among children in Ethiopia using MUAC and WHZ, along with the inflation factor (WHZ/MUAC). The prevalence of GAM varied across regions. In Tigray, MUAC (23.8%) was higher than WHZ (18.2%), whereas in Afar (MUAC 9.0%, WHZ 15.8%), Oromia (MUAC 6.0%, WHZ 6.7%), and South Ethiopia (MUAC 6.9%, WHZ 14.6%), MUAC underestimated GAM relative to WHZ. In Amhara (MUAC 13.2%, WHZ 10.6%) and Benishangul-Gumuz (MUAC 7.8%, WHZ 6.9%), MUAC slightly overestimated GAM. The largest differences in prevalence were observed in Somali (MUAC 2.8%, WHZ 15.8%) and Gambella (MUAC 2.4%, WHZ 9.6%), where MUAC identified only a small proportion of children classified as malnourished by WHZ..

**Table 2. Descriptive summary of WHZ and MUAC, by region, Ethiopia.**

| Regions | WHZ | | MUAC | | HAZ | | WAZ | | n |
|---|---|---|---|---|---|---|---|---|---|
| | mean | SD | mean | SD | mean | SD | mean | SD | |
| Tigray | -0.87 | 1.15 | 134.31 | 12.24 | -1.66 | 1.28 | -1.54 | 1.09 | 3033 |
| Afar | -0.97 | 1.09 | 141.53 | 12.82 | -1.55 | 1.68 | -1.54 | 1.19 | 1483 |
| Amhara | -0.75 | 1.04 | 138.72 | 12.02 | -1.73 | 1.24 | -1.49 | 1.01 | 3578 |
| Oromia | -0.48 | 1.07 | 143.36 | 12.42 | -1.67 | 1.47 | -1.28 | 1.07 | 4571 |
| Somali | -1.02 | 0.99 | 145.83 | 11.39 | -0.63 | 1.23 | -1.04 | 0.93 | 7341 |
| Benishangul-Gumuz | -0.68 | 1.01 | 141.22 | 11.62 | -1.86 | 1.36 | -1.52 | 1.08 | 537 |
| Gambella | -0.83 | 1.01 | 146.55 | 11.26 | -0.25 | 1.02 | -0.7 | 0.83 | 1140 |
| South Ethiopia | -0.76 | 1.18 | 141.31 | 12.47 | -0.88 | 1.35 | -1.01 | 1.04 | 1736 |
| Total | -0.81 | 1.07 | 142.09 | 12.6 | -1.22 | 1.43 | -1.24 | 1.05 | 23419 |

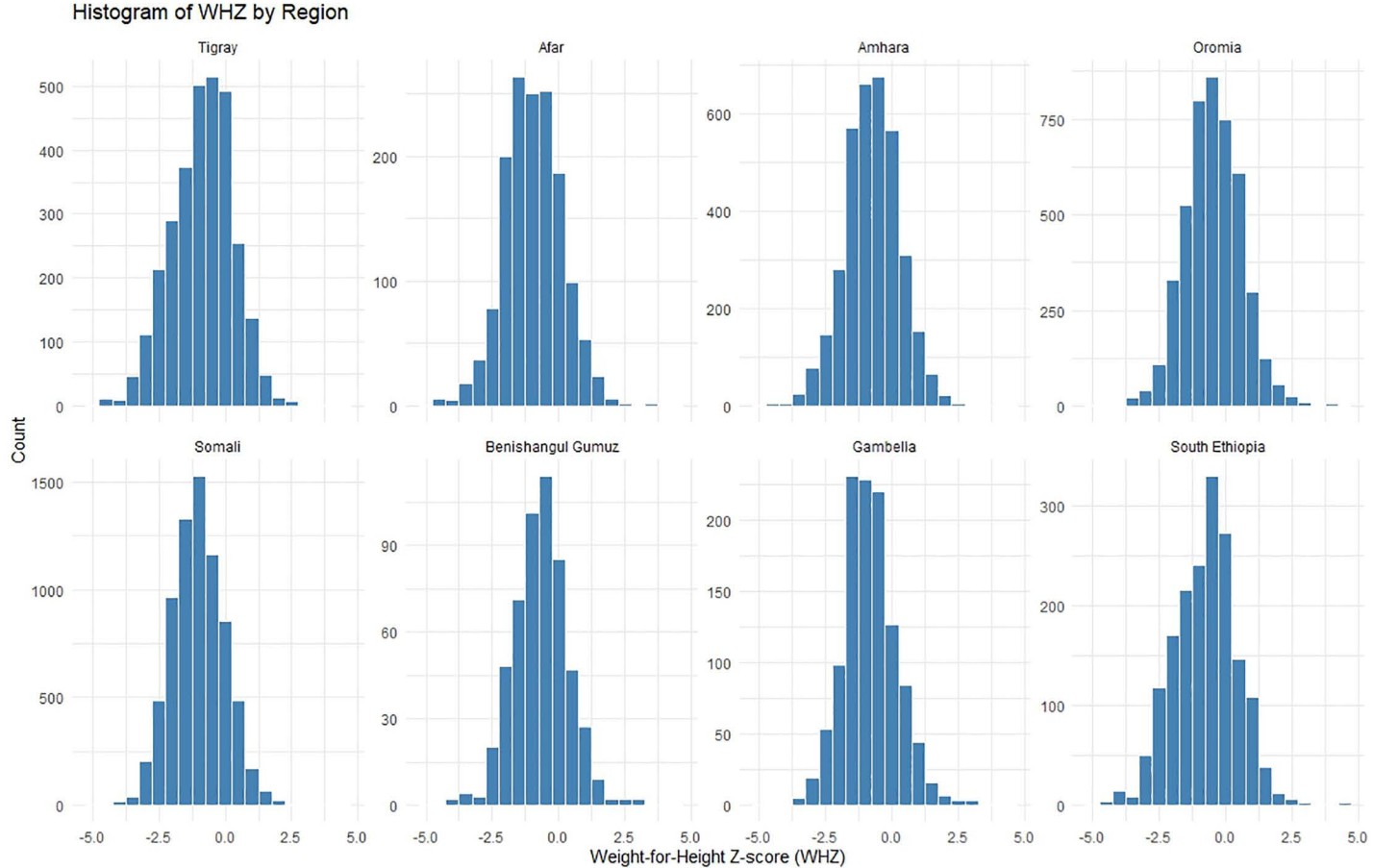

**Fig 1. Histograms of weight-for-height z-score (WHZ) by region, Ethiopia.**

Table 4 compares regional malnutrition classifications using WHZ and MUAC and shows the agreement between indicators using kappa statistics. Across regions, discrepancies existed between WHZ- and MUAC-based classification of children into MAM, Normal, and SAM categories. In Tigray, the distribution was similar between WHZ and MUAC, with 14.5% classified as MAM by WHZ versus 19.6% by MUAC, showing moderate agreement (kappa = 0.453). Afar and Somali showed low agreement, with kappa values of 0.211 and 0.128, respectively, indicating substantial differences in classification between the indicators. Amhara (kappa = 0.388) and Benishangul-Gumuz (0.365) showed fair agreement, while Oromia (0.260), Gambella (0.117), and Southern Ethiopia (0.325) had low to fair agreement. WHZ classified more children as MAM in Somali, Gambella, and Southern Ethiopia, whereas MUAC classified fewer children as malnourished in these regions. The kappa statistics showed that the concordance between WHZ and MUAC varied by region, with the highest agreement in Tigray.

**MUAC diagnostic performance**

Fig 4 shows the ROC curves assessing the diagnostic performance of MUAC in detecting wasting defined by WHZ in different regions of Ethiopia. The Area Under the Curve (AUC) values indicate overall accuracy. MUAC performed best in Benishangul-Gumuz (AUC = 0.892) and Tigray (0.865), showing high sensitivity and specificity. South Ethiopia (0.849),

**Fig 2. Histograms of MUAC by region, Ethiopia.**

Amhara (0.847), and Oromia (0.829) also exhibited good discriminative abilities, respectively. Afar (0.819), Gambella (0.807), and Somali (0.778) had comparatively lower AUCs, indicating moderate predictive performance. Overall, MUAC showed varying effectiveness in identifying wasted children across regions, performing better in most regions of northern and central Ethiopia than in some eastern and southern regions. Similarly, the accuracy of MUAC in identifying severe wasting cases across regions is generally a reliable tool for identifying severe wasting across regions, with variations in accuracy (Fig 5). Its diagnostic performance was high in Benishangul Gumuz (0.991) and Oromia (0.953), and slightly lower but still strong in Amhara (0.858) and Gambella (0.872).

### MUAC optimal threshold

The MUAC optimal threshold in detecting wasting based on WHZ varied across Ethiopian regions (Table 5). The optimal MUAC thresholds differed across regions, ranging from 126.5 mm in Tigray to 143.5 mm in Gambella. Sensitivity was lowest in Somali (0.70) and highest in Gambella (0.83), while specificity ranged from 0.66 in Gambella to 0.86 in Benishangul-Gumuz. This reflects MUAC's varying effectiveness in identifying wasted children as classified by WHZ across regions. The positive predictive values were generally low, highest in Tigray at 0.49, whereas the negative predictive values were consistently high across regions, exceeding 0.93 and reaching 0.98 in Oromia and Benishangul-Gumuz. The accuracy ranged from 0.67 in Gambella to 0.85 in Benishangul-Gumuz, with the highest Youden Index in

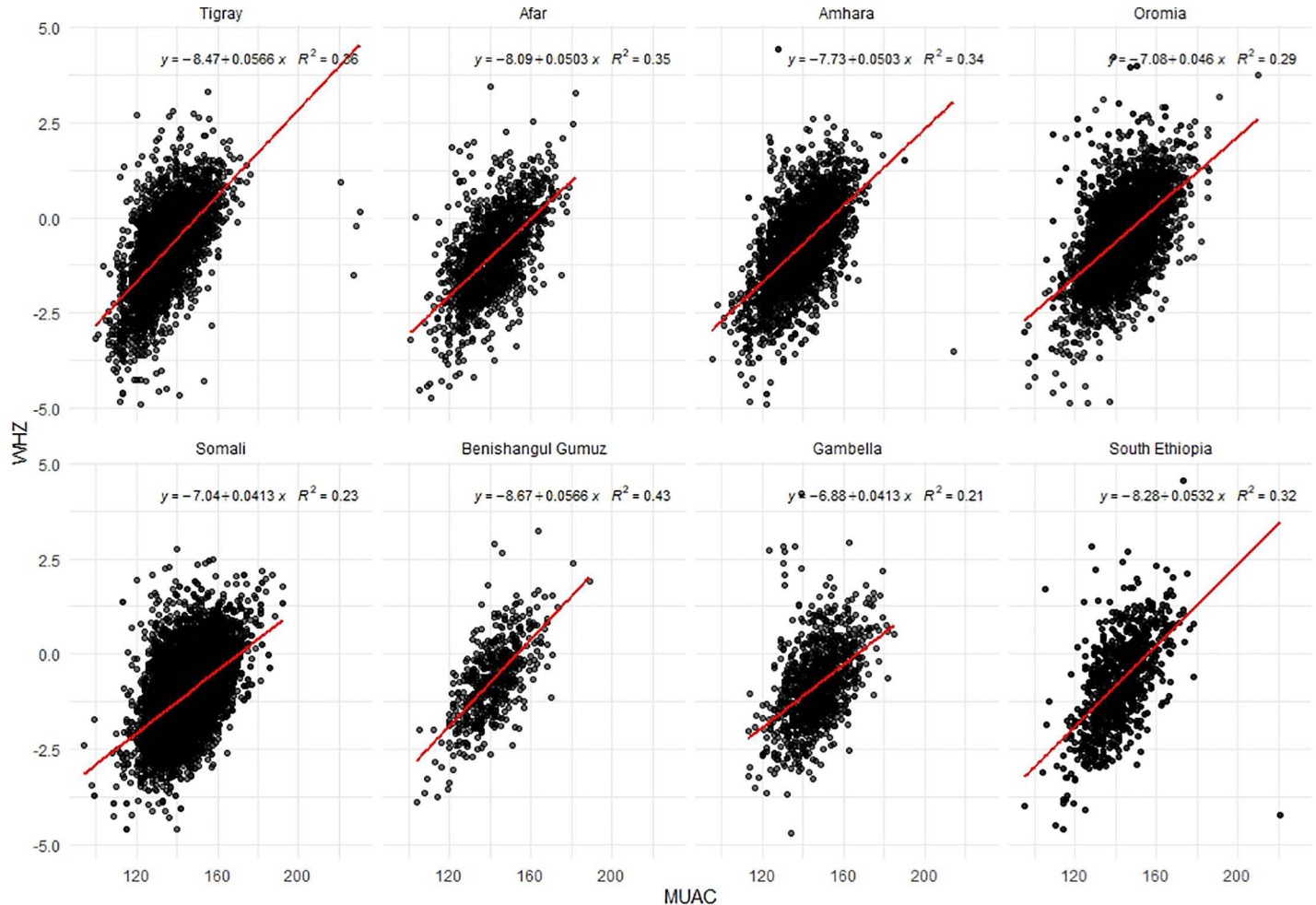

**Fig 3. Scatter plots for Mid-Upper Arm Circumference (MUAC) vs. Weight-for-Height Z-score (WHZ) by region, Ethiopia.**

**Table 3. Distribution of wasting by MUAC and WHZ by region, Ethiopia.**

| Region | MUAC | | WHZ | | Inflation factor (WHZ/MUAC) |
|---|---|---|---|---|---|
| | GAM (n) | GAM (%) | GAM (n) | GAM (%) | |
| Overall | 2000 | 8.5 | 3030 | 12.9 | 0.66 |
| Tigray | 721 | 23.8 | 552 | 18.2 | 1.31 |
| Afar | 134 | 9.0 | 235 | 15.8 | 0.57 |
| Amhara | 473 | 13.2 | 380 | 10.6 | 1.24 |
| Oromia | 276 | 6.0 | 305 | 6.7 | 0.89 |
| Somali | 207 | 2.8 | 1158 | 15.8 | 0.18 |
| Benishangul-Gumuz | 42 | 7.8 | 37 | 6.9 | 1.13 |
| Gambella | 27 | 2.4 | 109 | 9.6 | 0.25 |
| South Ethiopia | 120 | 6.9 | 254 | 14.6 | 0.47 |

**Table 4. Concordance between WHZ and MUAC varied region with Kappa statistics, Ethiopia.**

| Region | WHZ | | | MUAC | | | Kappa* |
|---|---|---|---|---|---|---|---|
| | MAM | Normal | SAM | MAM | Normal | SAM | |
| Tigray | 441 (14.5%) | 2481 (81.8%) | 111 (3.7%) | 595 (19.6%) | 2311 (76.2%) | 126 (4.2%) | 0.453 |
| Afar | 193 (13%) | 1248 (84.2%) | 42 (2.8%) | 116 (7.8%) | 1349 (91.9%) | 18 (1.2%) | 0.211 |
| Amhara | 318 (8.9%) | 3198 (89.4%) | 42 (1.7%) | 386 (10.8%) | 3103 (86.7%) | 87 (2.4%) | 0.388 |
| Oromia | 567 (5.7%) | 4266 (93.3%) | 45 (1%) | 213 (4.7%) | 4294 (93.9%) | 63 (1.4%) | 0.260 |
| Somali | 1031 (14%) | 6183 (84.2%) | 27 (1.7%) | 181 (2.5%) | 7127 (97.1%) | 26 (0.4%) | 0.128 |
| Benishangul-Gumuz | 30 (5.6%) | 503 (93.1%) | 7 (1.3%) | 35 (6.5%) | 495 (92.2%) | 7 (1.3%) | 0.365 |
| Gambella | 96 (8.4%) | 1031 (90.4%) | 13 (1.1%) | 23 (2%) | 1113 (97.6%) | 4 (0.4%) | 0.117 |
| South Ethiopia | 222 (12.8%) | 1482 (85.4%) | 32 (1.8%) | 96 (5.5%) | 1616 (93.1%) | 24 (1.4%) | 0.325 |

*Kappa values: ≤ 0 reflects no agreement; 0.01–0.20 suggest none to slight, 0.21–0.40 as fair agreement, 0.41– 0.60 represents a moderate level of agreement, 0.61–0.80 as substantial, and 0.81–1.00 signals an almost perfect agreement.

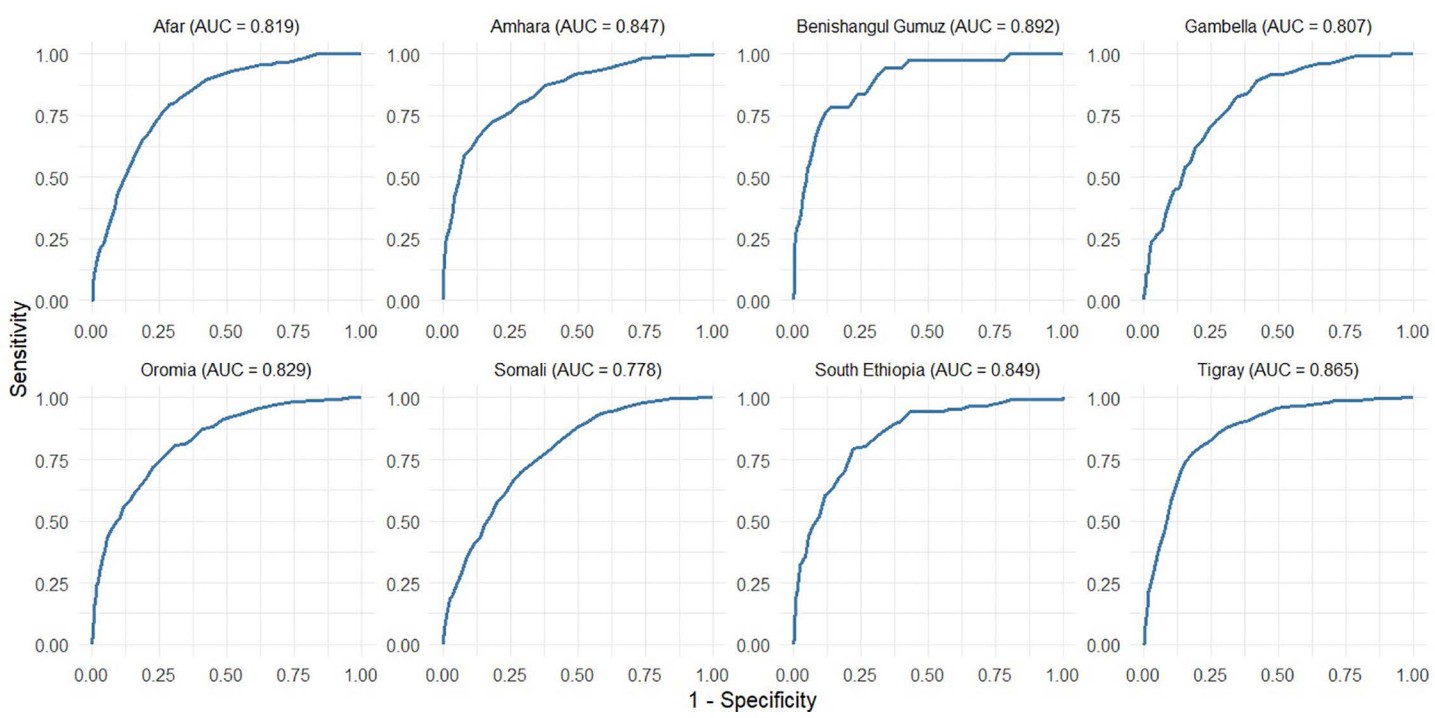

**Fig 4. Receiver's operating characteristic curve of MUAC for GAM against weight-for-height by region, Ethiopia.**

Benishangul-Gumuz (0.642) and the lowest in Somali (0.41), indicating the strongest performance in Benishangul-Gumuz and Tigray, while Somali showed weaker diagnostic efficiency.

Table 6 indicates that MUAC shows strong diagnostic performance for detecting severe wasting across regions, using WHZ as the gold standard. AUC values ranged from 0.858 in Amhara to 0.991 in Benishangul-Gumuz. Sensitivity and specificity were consistently high, and NPVs were nearly perfect (≈1.00), indicating that MUAC reliably ruled out non-severe cases. The Youden Index was highest in Benishangul-Gumuz (1.962 at MUAC 122.5 mm), showing excellent diagnostic performance at a low cutoff. In contrast, Amhara had the lowest index (1.588 at MUAC 130.5 mm), suggesting weaker utility despite a higher threshold.

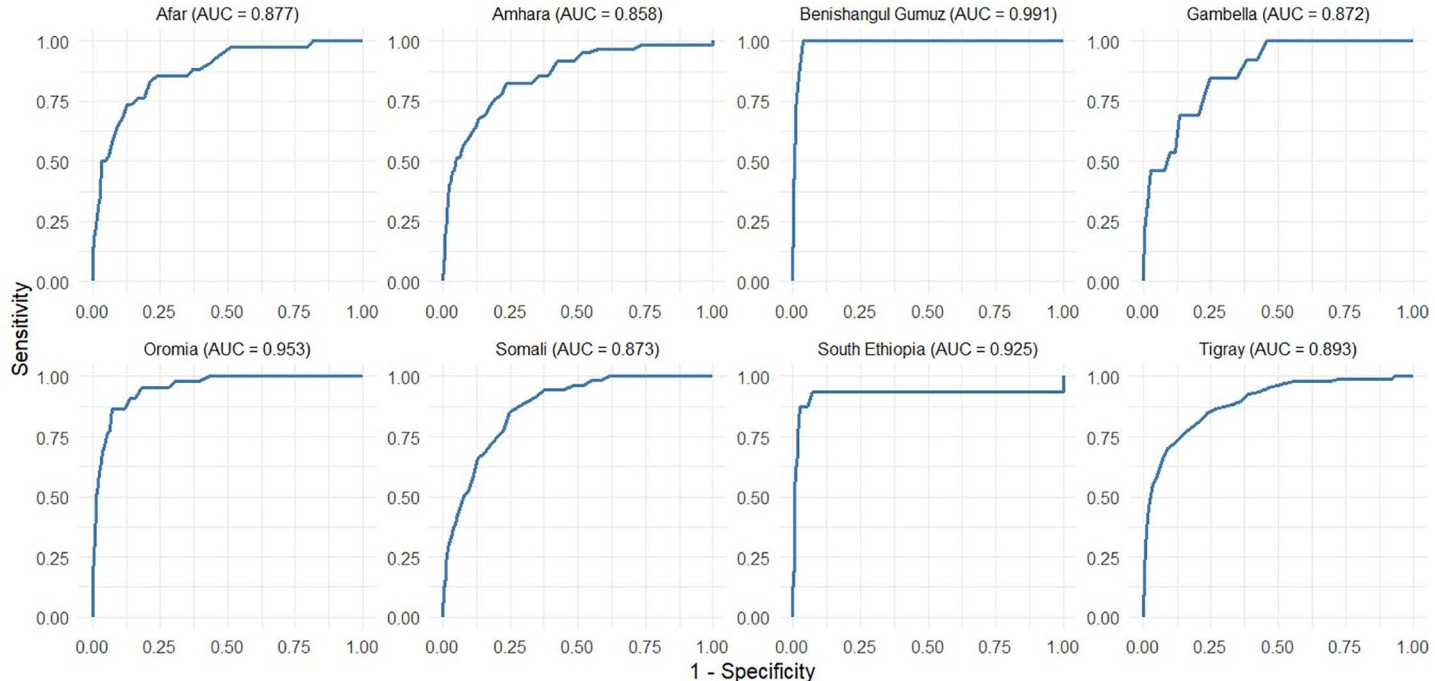

**Fig 5. Receiver's operating characteristic curve of MUAC for SAM against weight-for-height by region, Ethiopia.**

**Table 5. MUAC optimal threshold of wasting by region in Ethiopia, using WHZ as the Gold Standard.**

| Region | AUC | Threshold | Sensitivity | Specificity | PPV | NPV | Accuracy | Youden Index |
|---|---|---|---|---|---|---|---|---|
| Tigray | 0.865 | 126.5 | 0.78 | 0.82 | 0.49 | 0.94 | 0.81 | 0.595 |
| Afar | 0.819 | 136.5 | 0.80 | 0.71 | 0.34 | 0.95 | 0.73 | 0.510 |
| Amhara | 0.847 | 130.5 | 0.73 | 0.81 | 0.32 | 0.96 | 0.8 | 0.542 |
| Oromia | 0.829 | 138.5 | 0.81 | 0.69 | 0.16 | 0.98 | 0.7 | 0.496 |
| Somali | 0.778 | 141.5 | 0.70 | 0.71 | 0.31 | 0.93 | 0.71 | 0.410 |
| Benishangul-Gumuz | 0.892 | 131.5 | 0.78 | 0.86 | 0.29 | 0.98 | 0.85 | 0.642 |
| Gambella | 0.807 | 143.5 | 0.83 | 0.66 | 0.2 | 0.97 | 0.67 | 0.483 |
| South Ethiopia | 0.849 | 134.5 | 0.80 | 0.78 | 0.38 | 0.96 | 0.78 | 0.573 |
| All | 0.796 | 138.5 | 0.76 | 0.67 | 0.25 | 0.95 | 0.68 | 0.430 |

Note: The accuracy is driven by the true negatives.

## Optimal cut-offs

The diagnostic performance of MUAC in detecting wasting varied by age (Table 7), with WHZ serving as the gold standard. For children <24 months, low cutoffs (120–124 mm) had high specificity (0.92–0.97) but low sensitivity (0.31–0.51), with moderate PPVs (0.53–0.64) and high NPVs (0.89–0.91). Higher cutoffs (132–140 mm) increased sensitivity (0.83–0.96) but reduced specificity (0.40–0.45). NPVs remained high (>0.95), while PPVs fell below 0.25. The Youden Index peaked at 127–133 mm (0.528–0.545), identifying 128 mm as the optimal cutoff.

For children ≥24 months, the sensitivity was lower across the cutoffs. At 120–124 mm, sensitivity was 0.06–0.15 and specificity was 0.99, while higher cutoffs (136–145 mm) improved sensitivity (0.52–0.83) but reduced specificity to 0.57.

**Table 6. Diagnostic performance of MUAC for severe wasting by region in Ethiopia, using WHZ as the Gold Standard.**

| Region | AUC | Threshold | Sensitivity | Specificity | PPV | NPV | Accuracy | Youden Index |
|---|---|---|---|---|---|---|---|---|
| Tigray | 0.893 | 125.5 | 0.85 | 0.76 | 0.12 | 0.99 | 0.77 | 1.611 |
| Afar | 0.877 | 131.5 | 0.83 | 0.78 | 0.10 | 0.99 | 0.79 | 1.618 |
| Amhara | 0.858 | 130.5 | 0.82 | 0.77 | 0.06 | 1.00 | 0.77 | 1.588 |
| Oromia | 0.953 | 126.5 | 0.87 | 0.93 | 0.10 | 1.00 | 0.93 | 1.793 |
| Somali | 0.873 | 138.5 | 0.85 | 0.75 | 0.06 | 1.00 | 0.75 | 1.603 |
| Benishangul-Gumuz | 0.991 | 122.5 | 1.00 | 0.96 | 0.26 | 1.00 | 0.96 | 1.962 |
| Gambella | 0.872 | 139.5 | 0.85 | 0.75 | 0.04 | 1.00 | 0.75 | 1.599 |
| South Ethiopia | 0.925 | 125.5 | 0.94 | 0.93 | 0.19 | 1.00 | 0.93 | 1.865 |

**Table 7. Diagnostic performance metrics across various MUAC cutoffs by age category among children with wasting in Ethiopia, using WHZ as the Gold Standard.**

| < 24 months | | | | | | | ≥24 months | | | | | | |
|---|---|---|---|---|---|---|---|---|---|---|---|---|---|
| Cutoff | Sensitivity | Specificity | PPV | NPV | LR+ | LR− | Youden Index | Sensitivity | Specificity | PPV | NPV | LR+ | LR− | Youden Index |
| 120 | 0.31 | 0.97 | 0.64 | 0.89 | 10.08 | 0.71 | 0.278 | 0.06 | 1 | 0.63 | 0.89 | 12.77 | 0.94 | 0.058 |
| 121 | 0.36 | 0.96 | 0.59 | 0.89 | 8.23 | 0.67 | 0.313 | 0.08 | 0.99 | 0.65 | 0.89 | 13.51 | 0.93 | 0.072 |
| 122 | 0.39 | 0.95 | 0.57 | 0.9 | 7.64 | 0.64 | 0.343 | 0.09 | 0.99 | 0.62 | 0.89 | 11.78 | 0.91 | 0.085 |
| 123 | 0.45 | 0.94 | 0.55 | 0.91 | 7.04 | 0.59 | 0.382 | 0.11 | 0.99 | 0.58 | 0.89 | 10.2 | 0.9 | 0.103 |
| 124 | 0.51 | 0.92 | 0.53 | 0.91 | 6.29 | 0.53 | 0.429 | 0.15 | 0.99 | 0.57 | 0.89 | 9.68 | 0.87 | 0.13 |
| 125 | 0.59 | 0.89 | 0.5 | 0.93 | 5.59 | 0.46 | 0.487 | 0.19 | 0.98 | 0.55 | 0.9 | 9.02 | 0.82 | 0.172 |
| 126 | 0.64 | 0.87 | 0.47 | 0.93 | 5.09 | 0.41 | 0.514 | 0.21 | 0.97 | 0.53 | 0.9 | 8.14 | 0.81 | 0.183 |
| 127 | 0.68 | 0.85 | 0.44 | 0.94 | 4.53 | 0.38 | 0.53 | 0.24 | 0.97 | 0.5 | 0.9 | 7.41 | 0.79 | 0.205 |
| 128 | 0.72 | 0.82 | 0.42 | 0.94 | 4.05 | 0.34 | 0.545 | 0.26 | 0.96 | 0.49 | 0.91 | 6.94 | 0.77 | 0.224 |
| 129 | 0.75 | 0.79 | 0.39 | 0.95 | 3.59 | 0.32 | 0.54 | 0.29 | 0.95 | 0.46 | 0.91 | 6.16 | 0.75 | 0.24 |
| 130 | 0.78 | 0.76 | 0.36 | 0.95 | 3.22 | 0.29 | 0.538 | 0.31 | 0.94 | 0.43 | 0.91 | 5.58 | 0.73 | 0.258 |
| 131 | 0.81 | 0.73 | 0.34 | 0.96 | 2.98 | 0.26 | 0.536 | 0.35 | 0.93 | 0.42 | 0.91 | 5.24 | 0.69 | 0.285 |
| 132 | 0.83 | 0.69 | 0.32 | 0.96 | 2.73 | 0.24 | 0.528 | 0.38 | 0.92 | 0.38 | 0.92 | 4.51 | 0.68 | 0.295 |
| 133 | 0.87 | 0.66 | 0.31 | 0.97 | 2.55 | 0.2 | 0.528 | 0.41 | 0.9 | 0.35 | 0.92 | 4.02 | 0.65 | 0.31 |
| 134 | 0.89 | 0.63 | 0.3 | 0.97 | 2.38 | 0.18 | 0.516 | 0.45 | 0.88 | 0.34 | 0.92 | 3.74 | 0.63 | 0.327 |
| 135 | 0.92 | 0.59 | 0.28 | 0.98 | 2.24 | 0.13 | 0.511 | 0.48 | 0.86 | 0.32 | 0.92 | 3.5 | 0.6 | 0.346 |
| 136 | 0.93 | 0.55 | 0.27 | 0.98 | 2.05 | 0.13 | 0.475 | 0.52 | 0.84 | 0.31 | 0.93 | 3.25 | 0.57 | 0.359 |
| 137 | 0.94 | 0.51 | 0.25 | 0.98 | 1.93 | 0.12 | 0.452 | 0.57 | 0.82 | 0.3 | 0.93 | 3.12 | 0.52 | 0.388 |
| 138 | 0.95 | 0.48 | 0.24 | 0.98 | 1.83 | 0.1 | 0.431 | 0.6 | 0.79 | 0.28 | 0.94 | 2.92 | 0.5 | 0.397 |
| 139 | 0.96 | 0.45 | 0.23 | 0.98 | 1.73 | 0.1 | 0.402 | 0.65 | 0.77 | 0.28 | 0.94 | 2.8 | 0.46 | 0.416 |
| 140 | 0.96 | 0.4 | 0.22 | 0.98 | 1.61 | 0.1 | 0.365 | 0.68 | 0.74 | 0.27 | 0.94 | 2.65 | 0.43 | 0.425 |
| 145 | 0.99 | 0.25 | 0.19 | 0.99 | 1.32 | 0.06 | 0.237 | 0.83 | 0.57 | 0.21 | 0.96 | 1.95 | 0.3 | 0.403 |

NPVs remained high (≥0.89), and PPVs were modest (0.21–0.65), with the Youden Index peaking at 135–140 mm, with an optimal cutoff of 140 mm.

MUAC's diagnostic performance for severe wasting in Ethiopian children varies by age (Table 8). For children under 24 months, sensitivity increased from 0.21 at 111 mm to 0.98 at 137 mm, whereas specificity decreased from 0.99 to 0.46. PPV remained low (0.05–0.47), whereas NPV remained high (0.98–1.00), indicating an accurate classification of non-wasted children. Youden's index peaked at 125 mm (0.664), showing optimal sensitivity-specificity balance.

**Table 8. Diagnostic performance metrics across various MUAC cutoffs by age category in children with severe wasting in Ethiopia, using WHZ as the Gold Standard.**

| < 24 months | | | | | | | | ≥24 months | | | | | | | |
|---|---|---|---|---|---|---|---|---|---|---|---|---|---|---|---|
| Cutoff | Sensitivity | Specificity | PPV | NPV | LR+ | LR- | Youden Index | Cutoff | Sensitivity | Specificity | PPV | NPV | LR+ | LR- | Youden Index |
| 111 | 0.21 | 0.99 | 0.47 | 0.98 | 30.22 | 0.79 | 0.207 | 111 | 0.04 | 1.00 | 0.39 | 0.99 | 45.25 | 0.96 | 0.039 |
| 112 | 0.24 | 0.99 | 0.46 | 0.98 | 28.56 | 0.76 | 0.233 | 112 | 0.04 | 1.00 | 0.32 | 0.99 | 33.34 | 0.96 | 0.039 |
| 115 | 0.44 | 0.98 | 0.35 | 0.98 | 18.20 | 0.58 | 0.413 | 115 | 0.15 | 1.00 | 0.39 | 0.99 | 44.32 | 0.85 | 0.148 |
| 120 | 0.64 | 0.94 | 0.26 | 0.99 | 11.62 | 0.38 | 0.587 | 120 | 0.21 | 0.99 | 0.25 | 0.99 | 23.96 | 0.79 | 0.205 |
| 125 | 0.82 | 0.84 | 0.13 | 0.99 | 5.16 | 0.21 | 0.664 | 125 | 0.41 | 0.96 | 0.14 | 0.99 | 11.15 | 0.61 | 0.374 |
| 130 | 0.93 | 0.70 | 0.08 | 1.00 | 3.04 | 0.11 | 0.621 | 130 | 0.54 | 0.92 | 0.09 | 0.99 | 6.75 | 0.50 | 0.464 |
| 135 | 0.97 | 0.53 | 0.06 | 1.00 | 2.04 | 0.06 | 0.494 | 135 | 0.73 | 0.83 | 0.06 | 1.00 | 4.26 | 0.32 | 0.560 |
| 137 | 0.98 | 0.46 | 0.05 | 1.00 | 1.81 | 0.04 | 0.438 | 137 | 0.78 | 0.78 | 0.05 | 1.00 | 3.51 | 0.29 | 0.555 |

For children ≥24 months, the sensitivity increased from 0.04 at 111 mm to 0.78 at 137 mm, whereas the specificity decreased from 1.00 to 0.78. PPV remained low (0.05–0.39), NPV high (0.99–1.00). Youden's index peaked at 135 mm (0.560). Younger children (<24 months) were better identified at lower MUAC thresholds (125 mm), whereas older children required higher cutoffs (135 mm). Cutoffs of 125–135 mm provided an optimal sensitivity-specificity balance across age groups.

## MUAC misclassifications

Table 9 shows the false-positive rate (FPR) and false-negative rate (FNR) for MUAC at three cutoff points (<125 mm, <130 mm, and <140 mm) across Ethiopian regions, using WHZ as the gold standard. These findings show trade-offs between misclassification errors when applying different MUAC thresholds: At the <125 mm cutoff, false-positive rates (FPRs) were low, ranging from 1.2% in Somali to 13.4% in Tigray, indicating that few well-nourished children were misclassified as moderately wasted, based on WHZ classification (MAM-WHZ). However, FNRs were high, exceeding 40% in most regions and reaching 88.5% in Somali and 88.1% in Gambella. This cutoff would miss over two-thirds of nationally MAM-WHZ (66% overall), undermining its screening utility. At <130 mm, FPRs increased modestly, from 3.7% in Gambella to 27.9% in Tigray, as more children were incorrectly identified as MAM-WHZ. FNRs decreased substantially, falling to 13.8% in Tigray and 27.0% in Benishangul-Gumuz. Nationally, the average FNR dropped to 51.5%, meaning that half of

**Table 9. False-Positive Rate (FPR) and False-Negative Rate (FNR) across regions at various MUAC cutoff in estimating MAM-WHZ, Ethiopia.**

| Region | Cutoff | | | | | |
|---|---|---|---|---|---|---|
| | <125 | | <130 | | <140 | |
| | FPR | FNR | FPR | FNR | FPR | FNR |
| Tigray | 13.4% | 29.7% | 27.9% | 13.8% | 61.6% | 2.7% |
| Afar | 5.4% | 71.9% | 12.5% | 49.4% | 36.7% | 14.9% |
| Amhara | 7.8% | 41.3% | 17.0% | 29.2% | 48.1% | 8.7% |
| Oromia | 4.0% | 64.9% | 10.0% | 49.2% | 34.2% | 18.7% |
| Somale | 1.2% | 88.5% | 4.4% | 77.2% | 23.0% | 38.5% |
| Benishangul Gumuz | 4.8% | 51.4% | 10.6% | 27.0% | 40.2% | 5.4% |
| Gambella | 1.4% | 88.1% | 3.7% | 75.2% | 21.3% | 35.8% |
| South Ethiopia | 2.6% | 67.7% | 9.4% | 48.0% | 37.2% | 11.0% |
| Total | 4.8% | 66.0% | 11.4% | 51.5% | 36.2% | 21.6% |

the MAM-WHZ was missed, but fewer than at <125 mm. This cutoff reduced missed cases but falsely labeled one in ten well-nourished children as MAM-WHZ. At <140 mm, FPRs rose sharply, reaching 61.6% in Tigray and 48.1% in Amhara, meaning nearly half of well-nourished children would be misclassified MAM-WHZ. FNRs dropped significantly, with most regions below 20%, except for Somali (38.5%) and Gambella (35.8%). The national FNR was 21.6%, indicating that fewer than one in four MAM-WHZ were missed. However, the high national FPR of 36.2% means that over one-third of healthy children would be falsely identified as MAM-WHZ cases, potentially overburdening the supplementation programs.

Table 10 presents the FPR and FNR of MUAC across Ethiopian regions at cutoffs from 115 mm to 118 mm for estimating severe wasting based on WHZ (SAM-WHZ). The results show that increasing MUAC thresholds reduce missed cases (FNR) but increase false positives (FPR). At 115 mm, FPRs were low across regions, from 0.2% in Somali to 2.5% in Tigray, indicating that few well-nourished children were misclassified as having SAM-WHZ. However, FNRs were high, with most regions missing over half of the true SAM-WHZ cases. Somali and Gambella missed over 90% of SAM-WHZ cases (FNR = 92.1% and 92.3%), while Benishangul-Gumuz missed 42.9%. Nationally, the FNR was 70.8%, indicating that over two-thirds of the SAM-WHZ cases were undetected. At 116 mm, the FPRs increased slightly (to 0.3–2.8%), but the FNRs declined modestly. Tigray's FNR fell from 51.4% to 49.5%, whereas Oromia's dropped from 62.2% to 60.0%. Some regions maintained high SAM-WHZ misclassification rates: Somali (89.0%) and Gambella (92.3%). The national FNR was 67.9%. At 117 mm, FPRs increased slightly (up to 3.3% in Tigray), whereas FNRs decreased. Tigray reduced its FNR to 46.8%, Oromia to 55.6%, and southern Ethiopia to 37.5%. Nationally, the FNR dropped to 64.9%, but Somali (89.0%) and Gambella (84.6%) continued to miss most SAM-WHZ cases. At 118 mm, the FPRs increased marginally, with all regions below 4%. The FNRs showed greater improvement. Southern Ethiopia maintained the lowest FNR (37.5%), and Tigray reduced to 45.0%. The national FNR decreased to 63.3%. Somali (88.2%) and Gambella (84.6%) remained poor performers, missing nearly 9 in 10 SAM-WHZ cases.

## Discussion

Our analysis sought to investigate the diagnostic accuracy of MUAC for detecting wasting among Ethiopian children aged 6–59 months and explore the potential explanations for the discrepancies and their potential implications for nutrition program planning and design in Ethiopia. We used a large sample from SMART surveys conducted in eight regions of Ethiopia. Our findings indicate that the optimal MUAC threshold for children 6–59 months corresponding to WHZ < −2 was 139 mm, although it varied by age group and region. Younger children (<24 months) had a lower optimal cutoff (128 mm), whereas older children required a higher threshold (140 mm). Regionally, the thresholds ranged from 126.5 mm in Tigray to 143.5 mm in Gambella, indicating substantial geographic variation. We also observed discrepancies in GAM prevalence

**Table 10. False-Positive Rate (FPR) and False-Negative Rate (FNR) across regions at various MUAC cutoff in estimating SAM-WHZ, Ethiopia.**

| Region | Cutoff | | | | | | | |
|---|---|---|---|---|---|---|---|---|
| | 115 | | 116 | | 117 | | 118 | |
| | FPR | FNR | FPR | FNR | FPR | FNR | FPR | FNR |
| Tigray | 2.5% | 51.4% | 2.8% | 49.5% | 3.3% | 46.8% | 3.7% | 45.0% |
| Afar | 0.7% | 81.0% | 1.0% | 81.0% | 1.3% | 76.2% | 1.7% | 73.8% |
| Amhara | 1.8% | 64.5% | 2.2% | 61.3% | 2.4% | 59.7% | 2.7% | 59.7% |
| Oromia | 1.0% | 62.2% | 1.2% | 60.0% | 1.3% | 55.6% | 1.5% | 48.9% |
| Somale | 0.2% | 92.1% | 0.3% | 89.0% | 0.4% | 89.0% | 0.5% | 88.2% |
| Benishangul Gumuz | 0.6% | 42.9% | 0.6% | 42.9% | 0.9% | 42.9% | 0.9% | 42.9% |
| Gambella | 0.3% | 92.3% | 0.3% | 92.3% | 0.4% | 84.6% | 0.4% | 84.6% |
| South Ethiopia | 0.7% | 62.5% | 0.8% | 50.0% | 0.9% | 37.5% | 1.3% | 37.5% |
| Total | 1.0% | 70.8% | 1.2% | 67.9% | 1.4% | 64.9% | 1.6% | 63.3% |

by case definition: nationally, WHZ produced a higher estimate (12.9%) than MUAC (8.5%), with an average inflation factor of 1.31. However, this gap was not consistent; MUAC performed comparably or better in Tigray (inflation factor 1.31), but WHZ identified many more cases in Somali (inflation factor 0.18).

In surveys conducted in low- and middle-income countries (LMIC) and emergency settings, the correlation between MUAC and WHZ was moderate. For instance, Bilukha and Leidman analyzed 733 humanitarian surveys and identified a Spearman's ρ of approximately 0.55 (unadjusted $R^2 = 0.36$) when comparing the population prevalence of GAM as determined by WHZ and MUAC [20]. Similarly, Leidman et al. aggregated data from 882 representative surveys and reported a Pearson correlation coefficient (r) of approximately 0.49 ($R^2 = 0.24$) between individual MUAC and WHZ measurements in children aged 6–59 months [23]. These findings are consistent with the regional $R^2$ values observed in Ethiopia, which ranged from approximately 0.2 to 0.4. In essence, while a higher MUAC generally corresponds to a higher WHZ, there are numerous instances in which children are classified differently by each metric.

The patterns observed across regions were consistent with those of other studies. Bilukha et al. found that the MUAC and WHZ correlation was the strongest in the Middle East/North Africa and the weakest in Eastern/Southern Africa [20]. This aligns with the lowest correlations found in Somali/Gambella (Eastern Africa). Other African surveys have demonstrated similar moderate associations: a survey analysis in Mozambique reported a Spearman rank correlation coefficient of approximately 0.59 between individual MUAC and WHZ scores [23], comparable to the mid-range $R^2$ of the regions. Similarly, sub-Saharan-wide analyses revealed only moderate correlation; even adjusted models of country-level data showed $R^2 = 0.43$–0.50 [20]. In summary, global LMIC data confirmed a positive but not strong MUAC–WHZ link, which is consistent with our Ethiopian findings.

In our survey analysis, the GAM by WHZ was higher at 12.9% than 8.5% by MUAC, with an inflation factor of 0.66. This aligns with the general observation that WHZ often results in higher wasting rates than MUAC [20]. For instance, a recent survey in Ethiopia's Amhara Region reported a GAM of 13.2% by WHZ versus 8.6% by MUAC [31], reflecting the national trend. However, regional patterns show considerable variations. Pastoral and agro-pastoral regions, such as Somali, Gambella, Afar, and South Ethiopia, exhibited extreme inflation factors, with WHZ-based GAM rates far exceeding those of the MUAC. In contrast, highland agrarian regions, such as Tigray, Amhara, Oromia, and Benishangul, displayed much smaller differences or even slightly higher rates of MUAC. These variations underscore that the choice between MUAC and WHZ can significantly alter caseload estimates in certain areas, particularly pastoralist zones. The observed positive association between MUAC and WHZ across all regions aligns with the established understanding that both indicators reflect aspects of acute malnutrition. However, the variation in the strength of this association across regions underscores the complexity of nutritional assessment and the influence of contextual factors.

Analysis of 773 crisis surveys from humanitarian settings in 41 countries revealed a median GAM rate of 10.47% when assessed using WHZ, compared to 6.66% when measured using MUAC. In 74% of these surveys, the prevalence determined by WHZ exceeded that determined by MUAC [20]. This supports our broader conclusion that WHZ estimates are higher than MUAC. Numerous studies have highlighted significant discrepancies. For instance, in Cambodia, a GAM of 10.6% was reported by WHZ, whereas MUAC reported only 3.3% [32]. Similarly, a recent study in Somalia found a dramatic difference in wasting prevalence between the two measurements: 1.5% by MUAC versus 14.8% by WHZ [33]. Emergency surveillance data from Somalia also revealed marked inconsistencies between these two indicators [34]. Conversely, a study in southern Ethiopia found a 5.4% GAM by WHZ versus 10.5% by MUAC [35], while another study in Bangladesh found GAM prevalence by WHZ was 17.1% and by MUAC was 22.5% [36].

Several factors explain the variation in the MUAC vs WHZ relationship. Demographically, MUAC captures younger, smaller, and more stunted children (especially girls), whereas WHZ flags older and taller children [37]. These patterns can strengthen the MUAC–WHZ relationship: in regions with a higher proportion of young or stunted children (or more girls), WHZ tends to rise more rapidly with MUAC. This aligns with our findings (Table 2), in which northern regions with higher stunting levels showed stronger correlations. WHZ is highly sensitive to body proportions (leg length vs. trunk length).

Children in pastoralist or lowland groups often have longer legs (lower sitting-to-standing height ratios) than the global reference; this lowers their WHZ for a given weight and inflates the GAM by WHZ. In agrarian highland populations (higher sitting ratios), WHZ and MUAC tended to agree more [38]. Similarly, fat distribution matters: regions with the "thin-fat" phenotype (high central fat) may see WHZ affected differently than MUAC measurements [38,39]. In general, differences in body shape, stunting, and limb proportions drive much of the WHZ and MUAC gaps.

Relying on MUAC alone risks missing many at-risk children and underestimating the prevalence of wasting, thereby increasing mortality from undiagnosed cases [17,26,27]. Analysis of data from 48 countries estimated that a minimum of 300,000 annual deaths could occur among children excluded by MUAC-only screening [40]. Another multi-country pooled analysis demonstrated that MUAC and WHZ identify different child populations, with similar mortality risks for children missed by either measurement [41].

Our analysis also showed that higher MUAC thresholds detected more cases but also yielded more false positives. Increasing the MUAC to 130–136 mm doubled the sensitivity (49–67%) and the specificity was above 75%. Very high cutoffs (139–145 mm) gave sensitivities of 76–89%, but specificity dropped (47–67%), meaning most positives were not true wasted cases, although non-wasted children were correctly identified. Our findings are consistent with those of other studies that showed that the WHO MUAC 125 mm threshold has high specificity but low sensitivity [42,43]. Numerous studies suggest that raising the cutoff would identify many more children [36,44,45]. These studies consistently show that higher MUAC cutoffs would align better with WHZ cutoffs.

To further unpack these trade-offs, we examined age-specific patterns and regional differences in the misclassification errors. We observed age-related patterns in MUAC measurements. Children <24 months: Low MUAC cutoffs (120–124 mm) showed moderate specificity (0.92-0.97) but sensitivity under 50%. Raising MUAC to 130–132 mm increased sensitivity to 83% with a specificity of 70–75%. Peak Youden index was 127–133 mm (=0.53-0.55) for <2y children, suggesting 125 mm might be too strict for infants. Studies recommend a slightly higher MUAC for young children to detect more cases [10,36]. For children ≥24 months, sensitivity at 120–124 mm was low (6–15%) despite 0.99 specificity, missing most older wasted children. Higher MUAC improved sensitivity (52% at 136 mm, 83% at 145 mm) but reduced specificity to 0.57-0.64. Youden index peaked at 135–140 mm (=0.35-0.43). A single MUAC cutoff is inadequate for older children. Studies have confirmed lower MUAC sensitivity in older age groups [10,46]. Our findings support raising the MUAC cutoff for community screening, particularly for older children, to improve the sensitivity. This must be balanced against the lower PPV and resource needs.

Similar trade-offs were observed at the population level in different regions. Using MUAC <125 mm as a screening threshold keeps false positives very low (1–13% across regions); however, it misses the majority of children with acute malnutrition, more than two out of every three cases nationally (66%), and up to nearly 9 in 10 in Somali (88%). Raising the cutoff to MUAC <130 mm reduces the proportion of missed cases, with approximately half of malnourished children still not identified nationally (52%), although this comes at the cost of more false positives (up to 28% in Tigray). At MUAC <140 mm, the number of missed cases drops significantly to approximately one in five nationally (22%), meaning that most malnourished children are detected. However, the trade-off is a sharp increase in false positives, with over one in three healthy children misclassified (approximately 36% nationally and up to 62% in Tigray), which would place a heavy burden on treatment programs.

MUAC cutoffs can vary across populations, with age, gender, and geographic region influencing optimal diagnostic thresholds. Multiple studies show that a single, standardized MUAC cutoff fails to identify malnutrition accurately across contexts [10]. A study in Ethiopia revealed that optimal MUAC cutoffs ranged from 13.75 cm to 13.85 cm across ethnic regions, with sensitivity varying dramatically [47]. Similarly, another study conducted in Philippines found that while gender did not impact cutoffs, age substantially influenced optimal thresholds [17]. These variations underscore the need for population-specific nutritional screening approaches that account for local body composition, environmental conditions, and demographic characteristics to maintain diagnostic accuracy.

This study had several strengths and limitations. The dataset, drawn from the SMART+ aggregator, was based on large sample sizes and demonstrated good to excellent quality, supported by real-time data quality checks during collection. While data were available from eight regions, the study was not nationally representative. In addition, the use of SMART flags, which exclude measurements deemed statistically implausible, may have resulted in the omission of a small number of biologically plausible cases. However, sensitivity analysis showed that the impact of these exclusions on GAM prevalence by WHZ was minimal (12.9% without flags vs. 12.5% with flags), and the overall conclusions remain unchanged. Finally, although age and sex were included in our models, the datasets lacked other important factors that could influence the relationship between WHZ and MUAC.

## Implications for practice and policy

Given their partial overlap, MUAC and WHZ should be used together rather than as substitutes, especially in regions with low concordance [48]. This also supports the recommendation to consider region-specific MUAC cutoffs to better identify children missed by standard thresholds and ensure timely intervention for acute malnutrition. However, the feasibility of implementing such regional variations remains uncertain. Studies have also advocated for adjusting MUAC thresholds based on age, sex, and regional body composition norms to improve the diagnostic accuracy of some surveys [49]. Finally, nutrition programs and assessments should consider the local epidemiological context and measurement reliability when selecting screening tools for program design and setup. Second, the higher detection rate of MAM and SAM using WHZ suggests that relying solely on MUAC may underestimate the true burden of acute malnutrition. This has program and planning implications: using WHZ may increase caseloads and resource needs but ensures broader coverage. However, the WHO recommends the harmonized use of both indicators to avoid the exclusion of vulnerable children [50].

## Conclusion

Our analysis demonstrates that the current MUAC threshold of <125 mm misses the majority of malnourished children, particularly in Somali and Gambella, thus limiting its utility for screening at the community level. Cutoff selection is ultimately a policy decision that balances sensitivity (identifying at-risk children) and specificity (program capacity and burden). Although Somali and Gambella exhibited very high false-negative rates at the standard MUAC threshold, this does not imply that MUAC lacks programmatic value. MUAC remains an effective tool for identifying many severely malnourished children and is widely used for community-level screening due to its simplicity, low cost, and feasibility. The observed variation highlights the need for context-specific strategies and complementary approaches, rather than replacing MUAC. In these regions, relying solely on MUAC risks missing most wasted children, and programs may need to incorporate WHZ or combined criteria for case identification. Therefore, we recommend planning and resource allocation based on combined prevalence (WHZ < −2, MUAC < 12mm), particularly in Somali and Gambella, to ensure that all at-risk children are identified and supported in a timely manner.

## Acknowledgments

We extend our sincere appreciation to the Ethiopian Disaster Risk Management Commission (EDRMC), the Ethiopia Emergency Nutrition Coordination Unit (ENCU), the Nutrition Cluster Coordination Office, and all implementing partners for their dedicated teams for their invaluable support in data collection.

## Author contributions

**Conceptualization:** Alinoor Mohamed Farah, Aweke Kebede, Tafara Ndumiyana, Seifu Hagos Gebreyesus.

**Data curation:** Alinoor Mohamed Farah, Hamid Yimam Hassen, Sibhatu Biadgilign, Tesfamichael Awoke.

**Formal analysis:** Alinoor Mohamed Farah, Hamid Yimam Hassen.

**Investigation:** Alinoor Mohamed Farah.

**Methodology:** Alinoor Mohamed Farah.

**Project administration:** Beza Yilma, Seifu Hagos Gebreyesus.

**Supervision:** Seifu Hagos Gebreyesus.

**Writing – original draft:** Alinoor Mohamed Farah, Sibhatu Biadgilign.

**Writing – review & editing:** Alinoor Mohamed Farah, Hamid Yimam Hassen, Sibhatu Biadgilign, Aweke Kebede, Yakob Desalegn, Samson Gebremedhin, Kemeria Barsenga, Tafara Ndumiyana, Robert Ackatia-Armah, Helina Tufa, Firaol Bekele, Hailu Wondim, Eskeziaw Abebe, Seifu Hagos Gebreyesus.

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
