## [Decision Letter · Decision Letter 0]

26 Nov 2025

PGPH-D-25-02912

Improving case-detection of wasting among under-five-year-old children in Ethiopia: A secondary analysis of community based surveys in humanitarian settings

Dear Dr. Gebreyesus,

Thank you for submitting your manuscript to PLOS Global Public Health. After careful consideration, we feel that it has merit but does not fully meet PLOS Global Public Health’s publication criteria as it currently stands. Therefore, we invite you to submit a revised version of the manuscript that addresses the points raised during the review process.

Your manuscript has been assessed by three reviewers. While they indicate some interest in your work, numerous points have been raised which require your attention. Please consider these points carefully when preparing your revised manuscript and point-by-point response document.

We look forward to receiving your revised manuscript.

Kind regards,

Dr Jason Morgan

Staff Editor

Journal Requirements:

Please send a completed 'Competing Interests' statement, including any COIs declared by your co-authors. If you have no competing interests to declare, please state "The authors have declared that no competing interests exist".

Please amend your detailed Financial Disclosure statement. This is published with the article. It must therefore be completed in full sentences and contain the exact wording you wish to be published.If you did not receive any funding for this study, please simply state: “The authors received no specific funding for this work.”

Please provide separate figure files in .tif or .eps format and remove the embedded figures from the manuscript file.For more information about figure files please see our guidelines:https://journals.plos.org/globalpublichealth/s/figureshttps://journals.plos.org/globalpublichealth/s/figures#loc-file-requirements

In the online submission form, you indicated that “Data will be provided by the corresponding authors upon reasonable request.”All PLOS journals now require all data underlying the findings described in their manuscript to be freely available to other researchers, either1. In a public repository,2. Within the manuscript itself, or3. Uploaded as supplementary information.This policy applies to all data except where public deposition would breach compliance with the protocol approved by your research ethics board. If your data cannot be made publicly available for ethical or legal reasons (e.g., public availability would compromise patient privacy), please explain your reasons by return email and your exemption request will be escalated to the editor for approval. Your exemption request will be handled independently and will not hold up the peer review process, but will need to be resolved should your manuscript be accepted for publication. One of the Editorial team will then be in touch if there are any issues.

Additional Editor Comments (if provided):

Reviewers' comments:

Reviewer's Responses to Questions

**Comments to the Author**

1. Does this manuscript meet PLOS Global Public Health’s publication criteria ? Is the manuscript technically sound, and do the data support the conclusions? The manuscript must describe methodologically and ethically rigorous research with conclusions that are appropriately drawn based on the data presented.

Reviewer #1: Yes

Reviewer #2: Yes

Reviewer #3: Yes

2. Has the statistical analysis been performed appropriately and rigorously?

Reviewer #1: Yes

Reviewer #2: Yes

Reviewer #3: Yes

3. Have the authors made all data underlying the findings in their manuscript fully available (please refer to the Data Availability Statement at the start of the manuscript PDF file)?

Reviewer #1: Yes

Reviewer #2: Yes

Reviewer #3: No

4. Is the manuscript presented in an intelligible fashion and written in standard English?

Reviewer #1: Yes

Reviewer #2: Yes

Reviewer #3: Yes

Reviewer #1: WHO defines Wasting as WFH<-3SD score and or MUAC <12.5 cm .

Ideally MUAC performance should have been checked for whole GAM identified by both criteria . Authors have compared MUAC vs WFH .Please check References 19& 20 . They both are same.

Reviewer #2: This is well written and well conducted analysis that is relevant to an on-going discussion about the use of MUAC only programs in the community. Additionally, the Ethiopian ministry is very proactive about wasting management programs and so this paper may have policy impact. I do have a few major and minor critiques:

1) The paper’s framing and take homes are a bit confusing. You acknowledge that WHO recommends MUAC+WHZ to define SAM, and your conclusion is that this is correct. However the analysis looking at MUAC’s ability to predict WHZ, and you treat WHZ as the gold standard for SAM diagnosis. I think the whole results needs to be framed as MUAC ability to predict WHZ-SAM not as SAM generally. This is not incorrect in your discussion of false positives. In Table 10 you give the False positive rate of a MUAC cut off of 115mm for diagnosis SAM, but all children below 115mm have SAM! This actually only requires a small change – say WHZ-SAM not SAM alone. A similar issue can be raised about all your diagnostic accuracy statistics. For example, the sensitivity of MUAC to detect GAM should be MUAC+/(MUAC+ + WHZ+), but you are giving MUAC+/WHZ+. I agree with your approach, but I just think you need change the terminology your results to say WHZ-GAM and WHZ-SAM.

2) In the discussion, line 494, you say WHZ “overestimates prevalence compared to MUAC”. This is the opposite of the above issue – WHZ cannot overestimate because it is SAM. Both MUAC and WHZ are SAM, WHZ alone can only underestimate. Again, a small tweak is needed I think the word “overestimates” is the issue. I think that just saying WHZ estimates are higher than MUAC is sufficient.

3) Conclusion line 577, you say MUAC in two area had “extremely high false positive” even at normal cutoffs. That is not true because MUAC counts SAM. Again it is nuance in how you frame the results that could lead to people incorrectly thinking MUAC is of no value.

4) You mention in the introduction that discordant children, those who are MUAC+ WHZ- are still at risk. I think it is important to repeat this in the discussion. There is a risk that your analysis is misinterpreted to imply only WHZ matters. You do not conclude that, but it is important that readers are reminded that discordant children are still at risk.

5) The regional observation that you observe is very interesting and your interpretation, particularly the discussion of stunting and different body compositions is very insightful. One critique maybe that you’ve left room for readers to think that there may be regional variation in the accuracy of measurements – i.e. poor training in some regions or surveys. You only have one sentence in methods saying training was standardized. Is it possible to give more details to reassure readers this variation across regions is not about training differences?

6) You mention that the SMART flags exclude very plausible measurements in the methods and limitations. Can you not just turn those flags off as a sensitivity analysis?

7) You conclude that MUAC cutoffs should vary by region. Scientificall that might make sense, but programmatically is that a good idea? It sounds like a nightmare. I think like you should add a caveat that the programtic feasibility of such variation is unclear.

Minor

1) What happened to children with oedema in your analysis, were they excluded?

2) The “MUAC Diagnostic Performance” heading is repeat. This may be deliberate, but the sections do appear to be distinct.

Reviewer #3: I wish the authors all the best with their important work.

Feedback

Line 97: Add spacing between the statements and their corresponding references. Do this throughout the manuscript.

Line 112: mid-upper MUAC?

Line 113-114: Include the full reference.

Line 122-123: Explain why MUAC is more sensitive in detecting malnutrition among younger and female children compared with older and male children.

Line 136: Define GAM.

Line 151-153: Describe how children’s physical measurements were taken to ensure accuracy and consistency, including the types of anthropometric instruments used, their measurement precision and calibration.

Line 167: Report the number of missing or incomplete anthropometric records, if any were identified.

Line 178: Define HAZ and WAZ.

Line 207-208: Explain how Youden’s index was calculated and specify which index value was used to identify the optimal MUAC cutoffs (the higher YI?).

Line 207: Ensure that the 95% confidence interval for the AUC is included in the results section.

Line 208-210: Describe how sensitivity, specificity, positive predictive value, negative predictive value, accuracy, and likelihood ratios (LR positive and LR negative) were calculated/defined.

Line 244: In Table 1, add a separate row showing the total number of surveys and children instead of listing the total number of children under the Age (months) variable.

Line 217-219: Include information on participant consent and their rights during data collection.

Line 246: n(%)?

Figure 1-5: Remove the plot titles at the top of each figure because the titles are already provided in the captions.

Line 287-292: Do not repeat numbers that are already reported in the table.

Table 4: Specify in the table footnote what the Kappa statistic represents. Is it measuring agreement between MUAC and WHZ for identifying MAM, normal status, or SAM?

Line 337: Replace the repetitive subtitle with something like “MUAC Diagnostic Accuracy Measures.”

Line 449-450: Clarify whether the MUAC cutoff of 139 mm applies to the entire sample or only to older children.

Line 464-465: Explain what these numerous instances are by giving some examples of situations in which MUAC and WHZ classify children differently.

Line 467-475: Clarify whether the variation in MUAC–WHZ correlations across Ethiopian regions is due to differences in setting, population characteristics, dietary behaviors, or the fact that the two methods detect different aspects of body composition.

Line 487-489: Explain some of the key contextual factors that may influence the association between MUAC and WHZ.

Line 501-512: Excellent. Integrate these justifications into the earlier paragraphs discussing the correlation or association between the two methods.

**Do you want your identity to be public for this peer review?** For information about this choice, including consent withdrawal, please see our Privacy Policy .

Reviewer #1: No

Reviewer #2: No

Reviewer #3: No

---

## [Decision Letter · Decision Letter 1]

22 Jan 2026

Improving case-detection of wasting among under-five-year-old children in Ethiopia: A secondary analysis of community based surveys in humanitarian settings

PGPH-D-25-02912R1

Dear Dr Gebreyesus,

We are pleased to inform you that your manuscript 'Improving case-detection of wasting among under-five-year-old children in Ethiopia: A secondary analysis of community based surveys in humanitarian settings' has been provisionally accepted for publication in PLOS Global Public Health.

Best regards,

Julia Robinson

Executive Editor

Reviewer Comments (if any, and for reference):

Reviewer's Responses to Questions

**Comments to the Author**

Reviewer #2: All comments have been addressed

Reviewer #3: All comments have been addressed

publication criteria ? Is the manuscript technically sound, and do the data support the conclusions? The manuscript must describe methodologically and ethically rigorous research with conclusions that are appropriately drawn based on the data presented.

Reviewer #2: Yes

Reviewer #3: Yes

3. Has the statistical analysis been performed appropriately and rigorously?

Reviewer #2: Yes

Reviewer #3: Yes

4. Have the authors made all data underlying the findings in their manuscript fully available (please refer to the Data Availability Statement at the start of the manuscript PDF file)?

Reviewer #2: Yes

Reviewer #3: Yes

5. Is the manuscript presented in an intelligible fashion and written in standard English?

Reviewer #2: Yes

Reviewer #3: Yes

Reviewer #2: Thanks you for the changes to your manuscript. Good luck with the submission!

Reviewer #3: None

**Do you want your identity to be public for this peer review?** For information about this choice, including consent withdrawal, please see our Privacy Policy .

Reviewer #2: No

Reviewer #3: No
